# Identification and Fine Mapping of the Candidate Gene Controlling Multi-Inflorescence in *Brassica napus*

**DOI:** 10.3390/ijms23137244

**Published:** 2022-06-29

**Authors:** Hongchen Lu, Hanfei Wu, Guangfeng Zhu, Caijun Yin, Lun Zhao, Jing Wen, Bin Yi, Chaozhi Ma, Jinxing Tu, Tingdong Fu, Jinxiong Shen

**Affiliations:** National Center of Rapeseed Improvement in Wuhan, National Key Laboratory of Crop Genetic Improvement, College of Plant Science and Technology, Huazhong Agricultural University, Wuhan 430070, China; lhc1636671564@163.com (H.L.); HanFeiWu@webmail.hzau.edu (H.W.); ZhuGF1204@163.com (G.Z.); ycj0514dnaw@163.com (C.Y.); zhaolun@mail.hzau.edu.cn (L.Z.); wenjing@mail.hzau.edu.cn (J.W.); yibin@mail.hzau.edu.cn (B.Y.); yuanbeauty@mail.hzau.edu.cn (C.M.); tujx@mail.hzau.edu.cn (J.T.); futing@mail.hzau.edu.cn (T.F.)

**Keywords:** multi-inflorescence, *Brassica napus*, gene mapping, shoot apical meristem

## Abstract

As a desirable agricultural trait, multi-inflorescence (MI) fulfills the requirement of mechanized harvesting and yield increase in rapeseed (*Brassica napus* L.). However, the genetic mechanism underlying the multi-inflorescence trait remain poorly understood. We previously identified a difference of one pair of dominant genes between the two mapping parental materials. In this study, phenotype and expression analysis indicated that the imbalance of the *CLAVATA* (*CLV*)-*WUSCHEL* (*WUS*) feedback loop may contribute to the abnormal development of the shoot apical meristem (SAM). *BnaMI* was fine-mapped to a 55 kb genomic region combining with genotype and phenotype of 5768 BCF_1_ individuals using a traditional mapping approach. Through comparative and expression analyses, combined with the annotation in *Arabidopsis*, five genes in this interval were identified as candidate genes. The present findings may provide assistance in functional analysis of the mechanism associated with multi-inflorescence and yield increase in rapeseed.

## 1. Introduction

Rapeseed (*Brassica napus* L.), the second largest major oilseed crop worldwide, provides edible oil for the human diet as well as being a promising alternative source of protein for animal feed [1,2]. Conventional rapeseed crop management is costly, labor-intensive, and inefficient due to the low level of mechanization. To accelerate the adoption of mechanized cultivation and further yield increase, breeders proposed an ideal plant architecture as a guide for the development of new rapeseed varieties taking the “green revolution” in rice as a reference [3]. The compact plant architecture and the concentrated pod layer are essential in high-density planting and mechanized operation of rapeseed oil, which is urgently needed for rapeseed variety cultivation in the current agricultural management [4,5].

Plant architecture, mainly genetically determined, is defined as the three-dimensional structure [6,7]. Recent studies have shown that genes controlling plant architecture are involved in meristem regulation, hormone regulation, and other external processes [8,9]. The shoot apical meristem (SAM) can be divided into three functional zones: the central zone (CZ), peripheral zone (PZ), and rib zone (RZ) [10]. It provides continuous power for plant growth, and its maintenance and differentiation directly affect the subsequent morphological structure of plants [7,11]. Previous studies indicated that SAM is mainly regulated by the dynamic equilibrium of undifferentiated stem cell population. The negative feedback loop between the *CLAVATA* (*CLV*) and *WUSCHEL* (*WUS*) homeobox gene play a significant role during this process [12,13]. The mutations of the signaling peptide gene *CLV3* may cause meristems to enlarge due to stem cell over proliferation, leading to developmental defects in the morphology of inflorescence [13,14]. In *Arabidopsis thaliana*, RING domain ligase 1 (RGLG1) may act as ubiquitin ligases and the *CLV3*-*WUS* feedback loop-related genes were differential expressed in *rglg1 rglg2* double mutant plants, which exhibited increased lateral branches and altered phyllotaxy [15]. The deformed apical meristem led to the abnormal movement of plant inflorescence and stem [16,17]. Nevertheless, the molecular mechanism underlying shoot development in rapeseed was seldom concentrated.

Considering the important potential of plant architecture traits in genetic improvement of rapeseed, the research on genetic mechanism of plant architecture has attracted greater attention in recent years [18]. A semi-dwarf stature with narrow branch angles rapeseed mutant was characterized by ethyl methanesulfonate (EMS)-mutagenized, which would be valuable for breeding superior rapeseed hybrid cultivars and the yield increase in other Brassica crops [19]. Various Quantitative trait loci (QTL) for rapeseed branching were identified by using the traditional QTL-mapping [20,21]. The candidate genes for dwarf and compact plant architecture rapeseed were identified combined BSA-Seq based mapping and RNA-Seq profiling [22]. The main inflorescence structure, a key architecture component, is significant to increase yield of rapeseed [19,23]. Previous studies have shown that the number of pods on the main raceme (PR) have a significant contribution to the yield increase in rapeseed [24]. Integrating QTL mapping, microarray analysis, and whole-genome sequencing, a major QTL for multi-inflorescence was identified on A05 in *Brassica napus* [23]. A clavata-like phenotype mutant and the multi-stems mutant were reported separately, which greatly enriched the variation types of main stem in *Brassica napus* [25,26,27]. However, the genetic architecture associated with multi-inflorescence in rapeseed remains unclear.

In the present investigation, we characterized a new multi-inflorescence mutant in *B. napus* and identified the 55 K candidate interval of *MI* by using the traditional mapping approach. Furthermore, the candidate genes for the *MI* were analyzed via sequence comparisons and RT-PCR analysis. These results will be beneficial for marker-assisted breeding of multi-inflorescence rapeseed varieties and lay the foundation for further elucidation of the molecular mechanisms underlying multi-inflorescence in rapeseed.

## 2. Results

### 2.1. Morphological of Multi-Inflorescence and Possibility of Yield Increase

Compared with wild-type L780 parents, *MI* mutant has parallel main inflorescences (Figure 1) and the dividing nodes on the main stem are clearly visible. At the vegetative development stage, seedlings of *MI* mutant have more rosette leaves than wild-type L780 (Figure 1f and Appendix A). At the flowering stage, *MI* mutant exhibited lower plant height than that in L780 (Appendix A), and significant higher number of main inflorescence pods at maturity (Figure 1b). Moreover, the agronomic characters of single inflorescence and multi-inflorescence plants in BC_4_F_1_ population were measured, respectively. The proportion of main inflorescence yield is between 24.77% and 92.19%, and the range of single inflorescence is between 15.59% and 33.92%. The plant with the multiple inflorescences has more pods, while the pod length and number of grains per pod have little difference (Appendix A). The multi-inflorescence traits therefore undoubtedly will provide the possibility of mechanized operation and yield increase.

To explore whether the difference in main inflorescence was caused by a change in meristem morphology or size, the SAM of L780 and *MI* were observed by histological observation. Stereomicroscope observation illustrated that the number of SAM in *MI* was more than that in the wild-type SAM at initial stage of inflorescence meristem (Figure 1c,d). Increased SAM led to more leaf primordia and flower meristem (Figure 1f). Consequently, the abnormal apical development could be the immediate reason of the formation of multi-inflorescence traits (Figure 1c,d). Subsequently, to prove the hypothesis, the transcription level of SAM development-related genes in L780 and *MI* were investigated by real-time RT−PCR. The result indicated that several genes were significantly changed, including *CLV1*, *CLV2*, *CLV3*, and *WUS* (Figure 2). Compared with wild type, the transcription level of *CLV2* had increased significantly and the opposite results were observed in the expression of *WUS*. The transcription levels of *CLV1* in *MI* were approximately half of that in L780. The data was consistent with the result of stereomicroscope observation, which implied the natural variation of SAM in *MI* is caused by *CLV*-*WUS* feedback loop.

### 2.2. Fine Mapping of BnaMI Gene

In previous study, *BnaMI* gene was mapped at the physical interval of 2.5 Mb on chromosome A05 of *B. napus* using *Brassica* 60K SNP BeadChip Array combined with molecular marker technology, L04−2 and L06−14 were the closest flanking markers [28]. To reveal the molecular mechanisms of the multi-inflorescence phenotype, simple-sequence repeat (SSR) markers were designed and *BnaMI* was further located on chromosome A05, an interval between markers S56 and S86 (Figure 3a), and corresponding to a 652 kb region of *B. napus* Darmor-*bzh* physical map. Using a larger BC_4_F_1_ population of 3487 plants, *BnaMI* was then fine-mapped to a 55 kb genomic region between intron polymorphism (IP) markers IP-29 and IP-57; two and four recombination events were detected for IP−29 and IP−57, respectively (Figure 3). The frequency distribution on multi-inflorescence traits in BC_4_F_1_ population was exhibited in Appendix A. 

### 2.3. Candidate Gene Analysis in the Mapping Region

In 55 kb region, 14 genes were annotated or predicted (Appendix A) in *Arabidopsis*. For each of the 14 genes, we comparatively sequenced the genomic fragments covering the promoter region and the complete coding region from two mapping parents. The results showed that six genes showed amino acid sequence variation (Table 1). The nucleotide sequence alignment of the candidate genes, including the promoter region and the complete coding region, were provided in Appendix A, respectively.

Among them, *BnaA05g08870D* was similar to *AT2G35035*, a urease accessory protein D, of which the analog was associated with tillers in rice [29], increasing the probability to be a candidate gene. *BnaA05g08870D* has 19 SNPs in the coding region, including 14 nonsynonymous mutations, and four amino acid variation occurs in the conserved domain between L780 and *MI* mutant. At the same time, the deletion of two bases resulted in frameshift and the early termination of gene translation (Figure 4a). It is worth noting that two candidates *BnaA05g08900D* and *BnaA05g08910D*, of which the analogs belong to RING/U-box superfamily protein in *Arabidopsis*. Among them, *BnaA05g08900D* encode an E3 ligase-like protein, the biological function of the other gene remains unclear [30]. *BnaA05g08900D* has 24 SNPs in the coding region, including 18 nonsynonymous mutations, with seven amino acid variation occurring in *MI* mutant (Figure 4a). *BnaA05g08900D* has seven SNPs in the coding region, which cause four amino acid variations. Therefore, these two genes could be the candidates controlling multi-inflorescence trait based on previous observations of meristem. In addition, the expression levels of these genes in SAM were detected (Figure 4b). The qRT−PCR results show that the transcript levels of *BnaA05g08870D* was significantly higher in *MI* than in L780, which was consistent with *MI* mutant dominant characteristics.

### 2.4. An Identified 8.5 kb Insertion by Comparative Sequencing

When comparing the promoter region these 14 genes between L780 and *MI* mutant, a 8.5 kb insertion in the regulatory region 506 bp upstream of *BnaA05g08900D* was identified by Sanger sequencing (Figure 5a,b). The obtained sequence was listed in Appendix A. The identified 8.5 kb sequence was then blasted in NCBI (https://blast.ncbi.nlm.nih.gov/Blast.cgi, last accessed on 6 April 2022) and the alignment result indicated that this sequence come from A05 chromosome of *Brassica rapa* and it contains two genes *BraA05t20012Z* (*ORF12*) and *BraA05t20013Z* (*ORF13*) (Figure 5c).

To confirm the transcriptional activity of these two genes, three and two pairs of primers were designed between exons, respectively. The same primer can amplify bands in *MI* mutant, but not in L780 (Figure 5c). The protein sequence of these two genes were blasted in *Arabidopsis* database (https://www.arabidopsis.org/Blast/index.jsp, last accessed on 6 April 2022), the alignment result demonstrated that *ORF13* were highly similar to the *AT5G28250*, a transposable element gene similar to *Ulp1* protease family protein and no more information about *ORF12* was available. In addition, we analyzed their transcriptional expression levels between L780 and *MI*. The transcript levels of *BraA05t20012Z* and *BraA05t20013Z* were significantly up-regulated expression in SAM of mapping parent *MI* (Figure 6a), which is consistent with the dominance characteristic of multi-inflorescence. The expression patterns analysis of *ORF12* and *ORF13* were performed at the different part of *MI.* According to the results, the expression level of *ORF12* was low in the roots, stems, leaf, bud and silique, but high in flower and shoot apex (Figure 6b). Similarly, the high expression levels in shoot apex occurred on *ORF13* (Figure 6c). The data obtained may be related to the cause of *MI* traits formation, suggesting these two genes may be prioritized as candidate genes of *MI*. In brief, through sequence and expression analysis, combined with the annotation in *Arabidopsis*, five genes (*BnaA05g08870D*, *BnaA05g08900D*, *BnaA05g08910D*, *ORF12*, and *ORF13*) were identified as promising candidates for *BnaMI*.

## 3. Discussion

Rapeseed is an important oilseed crop worldwide. The ideal plant architecture of rapeseed is closely related to the yield, which has always been a popular topic among breeders. Limited by planting costs and labor reduction, the cultivation of rape seed oil varieties suitable for mechanized harvest has been promoted to a strategic position, apart from improved oil content and seed yield of rapeseed [5]. Continuous exploration has been made around the key factors. The petal free type, the pod layer with canopy type, and compact branches can be identified as the ideal plant types of rapeseed, with great potential for higher yield. In recent years, plant architecture related mutants have been reported in *Brassica napus* [23,25,26,27]. Previous studies have shown that multi-inflorescence *MI* is a new type of rapeseed material (Figure 1), which could speed up the process of mechanization [28]. In this study, the candidate interval for multi-inflorescence was narrowed to an approximately 55 kb region on the basis of the flanking markers, localized in 5.38–5.44 Mb region of chromosome A05 according to the reference genome ‘Darmor-*bzh*’. Breeding for high yield is a complex and long-term process, and the closely linked markers developed in our study could accelerate the breeding process of multi-inflorescence varieties.

The plant architecture is closely related to the development of apical meristem [8] and the interaction between *WUS* and *CLV* maintains the normal development of plant apical meristem [31,32]. The molecular mechanism underlying shoot development are well known in *Arabidopsis*. *WUS* mutants generally lead to the failure of normal maintenance of apical meristem and *CLV* mutants generally show the increase in meristem and the deformity of inflorescence meristem [33,34,35]. However, the role of SAM morphology underlying shoot development in rapeseed is less appreciated. In this study, phenotype and histological analysis revealed that extremely increased SAM and IM led to the *MI* phenotype, parallel growth branches and increased main inflorescences (Figure 1). This result distinguished the previous observation in *dt* mutant of *Brassica napus* [27]. Moreover, the transcription level of SAM development-related genes in L780 and *MI* exhibited differential expression, *WUS* and *CLV* showed opposite expression particularly (Figure 2). The data indicated that genes involved in SAM development could have interfere with the *CLV3*-*WUS* feedback loop, resulting the abnormal development of *MI*. This research of *MI* materials provided better understanding of the relationship between apical development and plant architecture in *Brassica napus*.

The development of plant apical meristem is a complex process, regulated by multiple gene feedback and affected by various hormones [36]. *LOG* encodes a cytokinin-activating enzyme which works in bioactive cytokinin synthesis [37]. Cytokinins mediate stem cell size through *WUS* expression; *CKX5* and *CKX6* repress *WUS* expression via the degradation of cytokinin [38]. The genes involved in cytokinin biosynthesis and signal transduction were altered significantly in *dt* mutant, which provide an explanation for the irregular development of SAM [27]. *ARF5*, an ARF transcription factor, plays a key role in specifying meristematic and primordium fate [39], which directly represses *ARR7*/*ARR15* and activates *AHP6* through the regulation of cytokinin homeostasis [40]. In addition, *ARF5* may directly activates the floral meristem identity gene *LEAFY* (*LFY*) by binding the promoter region, which regulated inflorescence architecture [41]. Recent reports in *Arabidopsis* demonstrated that the interaction between the auxin pathway and *LFY* mediate the initiation and specification of FMs [42,43]. Nevertheless, the candidate genes are not associated with these phytohormone related genes. Instead, *BnaA05g08900D* encoding an E3 ligase-like protein, and *BnaA05g08910D* associated with RING/U-box superfamily protein, were involved in candidate region. Previous studies have pointed out that RINGfinger protein and zinc finger protein played vital roles in shoot apical meristem maintenance and floral organ identity [30,44]. *BnaA05g08870D*, encoding urease accessory protein D (UreD), is necessary in vivo urease activation [45]. Compared with wild type, the *Osured* mutant line exhibited obvious growth inhibition with reduction in tiller numbers and urease activity at seeding and tillering stage of rice [29]. In our research, amino acid variation and differences in expression in *BnaA05g08870D* were simultaneously observed between *MI* and L780 (Figure 4). Thus, it can be speculated that these genes may be involved in development of multi-inflorescences in *MI*.

When the comparative sequencing was performed between the two mapping material, a transposable element gene insertion in the promoter region of *BnaA05g08900D* was detected in *MI* (Figure 5a), which is highly expressed in SAM and stem (Figure 6c). Transposable elements have tremendous effects on genome structure and gene function, and a few active elements causing the genomic alterations may have major outcomes for a species [46]. Previous studies demonstrated that transposon mediated maize domestication, epigenetic regulation of rice, and morphological variation of tomato fruit [47,48,49]. Much evidence has demonstrated that insertions of TEs were related to yield traits. For instance, introgression of alleles lacking the insertions of two transposable elements in the regulatory region of *KNR6* led to sharp yield increase in maize [50]. A 3.7 kb transposable element insertion stimulated high gene expression, which is associated with silique elongation and seed enlargement in rapeseed [51]. In consideration of the dominance characteristic of multi-inflorescence, these two inserted genes should be included although the expression level of *BnaA05g08900D* showed no difference between two mapping parents.

In conclusion, the multi-inflorescence gene was fine-mapped to 55 kb region using a traditional mapping approach. Sequence and expression were analyzed combined with of the reported function in *Arabidopsis*; the result implied that five genes were prioritized as promising candidates for *BnaMI*. The genetic transformation in rapeseed is urgent for further functional analysis. Our research deepened the understanding of shoot development in *B. napus* and laid a foundation for functional analysis of the mechanism associated with multi-inflorescence and yield increase in rapeseed.

## 4. Materials and Methods

### 4.1. Plant Materials, Population Development, and Trait Evaluation

The materials *MI* and L780 were self-crossed for more than six generations, and are provided by Department of Rapeseed Research of Huazhong Agricultural University. BC_2_F_1_ and BC_3_F_1_ generations were built using backcross method. Recombinant individuals were screened from BC_3_F_1_ population to the built BC_4_F_1_ population, which were used for fine mapping *MI*. All of these populations were planted at the experimental station of Huazhong Agricultural University, Wuhan, China. The planting density between rows was 20 cm and the row spacing was 25 cm. Regular field management was conducted according to local agricultural practices. For most experiments, as well as for sequence and expression analysis, the sequenced inbred lines ZS11 and *B. napus* Darmor-*bzh* were used [52,53].

For the investigation of multi-inflorescence characteristics, the phenotype of individual plants in each population were observed after bolting of rapeseed in March. The phenotype of each individual plant was examined more than twice. For the evaluation of all agronomic traits, individual plants with the same growth status and free of disease were selected for phenotypic investigation and analysis. The BN refers to the number of effective branches arising from the main shoot. The length of the main inflorescence (LMI) was measured from the base of the highest primary effective branch to the tip of the main inflorescence [20]. The total number of seeds on the main inflorescence was used to measure the thousand-seed weight (TSW), by weighing 500–1000 seeds and then converting the weight into TSW as previously described [54].

### 4.2. Histological Analysis

For histological analysis of SAM, 10 plants were sampled from L780 and *MI* every two days, respectively, until the end of flower bud differentiation. At the early stage of SAM development, all distinguished leaves of the sampled plants were removed until they were difficult to distinguish with the naked eye. The stripped SAMs from plants were carefully peeled off with an anatomic needle under a stereomicroscope (NikonSMZ25). Finally, they were photographed by using a software that was attached to the stereomicroscope [55].

### 4.3. Genomic DNA Extraction and Marker Development

The young leaves were used to extract genomic DNA by a modified CTAB method [56]. After measuring the concentration of DNA solution with NanoDrop 2000, dilute the solution concentration to 50 ng/μL. Most of the markers used during fine-mapping in this study were SSR markers and IP marker primers. The SSR markers were batch designed by using bioinformatics methods utilizing the reference genome sequence of B. napus Darmor-*bzh* genome [57] with the exception of several primers developed by web-based SSR finder tool (https://bioinfo.inf.ufg.br/websat/, last accessed on 6 April 2022). SnapGene was used to design IP marker primers within the conserved sequences on both sides of introns. According to the field inflorescence phenotype observation, two multi-inflorescence bulks (BD1 and BD2) and two single inflorescence bulks (BS1 and BS2) were built for SSR and IP marker screening. Each pool containing 10 individuals were selected randomly from BC_1_F_1_ population. The genotype identified by polymorphic markers combined with phenotype pointed out the recombinant individuals to promote narrowing the candidate interval of *MI*. The general scar program was used for gene fragment amplification and the amplified products were detected on 1.2% agarose gel. SSR amplified products were then separated on 1% polyacrylamide denaturing sequencing gel and shown by silver nitrate staining. Linkage analysis was performed using JoinMap4. The developed markers in this study were listed in Appendix A. 

### 4.4. Sequencing and functions prediction of candidate genes

The function predictions of candidate genes were adopted from their orthologs in *A. thaliana*, using BLAST tool of TAIR website (https://www.arabidopsis.org/, last accessed on 6 April 2022). The gene specific primer pairs were designed according to the nucleotide sequence of candidates (Appendix A) and used to amplify the genomic DNA sequence between the mutant *MI* and the wild-type L780. The purified products were ligated into pTOPO-Blunt Simple Vector (Aidlab Biotech, Beijing, China) following the manufacturer’s instructions and then transformed into *Escherichia coli* DH5α cells (Tsingke, Wuhan, China). Some long fragments were cloned into pCAMBIA2300 vector by using single restriction endonucleases (*Pst* I) and a ClonExpress II One Step Cloning Kit (Vazyme, Nanjing, China). Each sequence had at least two biological repeats. The positive clones were sequenced by Tsingke Biological Company (Wuhan Co., Ltd., Wuhan, China). The obtained sequence of candidate genes between two mapping parents were converted to acid sequence for further analysis using MEGA-X software.

### 4.5. RNA Extraction and Quantitative Real Time PCR (qRT−PCR)

Different tissues were sampled for total RNA extraction, using RNAprep Kit (Tiangen, Beijing, China), including the root, stem, leaf, flower, bud, silique, and shoot apex of L780 and *MI*. The samples were collected and directly frozen in liquid nitrogen and stored at −80 °C for RNA extraction. Three biological replicates were used for all samples. Total RNA of these materials were extracted by using a Plant RNA Kit following the manufacturer’s instructions (TIANGEN Biotech, Beijing, China) and were stored at −80 °C. A 2 µg total RNA was reverse transcribed to cDNA, following the manufacturer’s instructions (TaKaRa, Dalian, China). For real−time RT−PCR analysis, cDNA was diluted 1:50 with sterile water, and the reactions were performed as described by the manufacturer. qPCR was performed using 10 μL SYBR Green Real-time PCR Master Mix (Toyobo Life Science, Osaka, Japan) in a CFX96 instrument (Bio-Rad, CA, USA). The whole experiment operation was conducted in a dim room. The real-time PCR program was 5 min at 95 °C; followed by 40 cycles of 15 s at 95 °C and 20 s at 56 °C and 30 s at 72 °C; then 10 s at 95 °C and 65–95 °C, increment 0.5 °C every 5 s for melting curve detection. Triplicate quantitative PCR experiments were performed for each gene. Quantitative RT−PCR measurements were obtained by using the relative quantification 2^−∆∆Ct^ method [58]. Data are presented as the mean of three biological replicates ±SD. The average relative expression levels were calculated and t tests were performed to determine the significance of differences in expression levels of different genotypes. The data were used for GraphPad Prism 8 (https://www.graphpad-prism.cn/, last accessed on 6 April 2022) to form a histogram. All gene specific primers used for the amplification are listed in Appendix A, and the expression values obtained were normalized against actin gene.

## Figures and Tables

**Figure 1 ijms-23-07244-f001:**
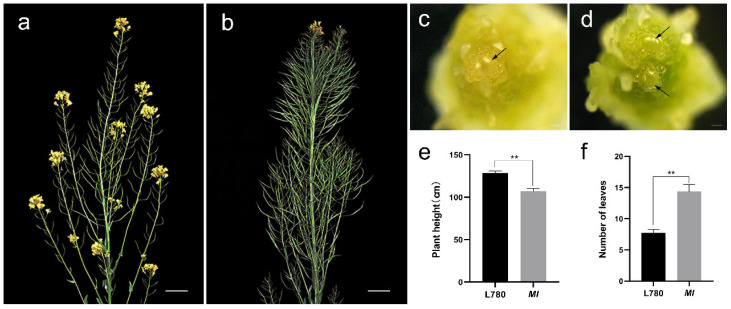
Multi inflorescence-related traits and morphology of shoot apical meristems (SAMs) in *MI*. (**a**,**b**) Plant morphology of L780 (**a**) and *MI* (**b**); bar = 10 cm. Inflorescence meristems (IM) of L780 (**c**) and *MI* (**d**) in initial stage of IM differentiation. Black arrows indicate SAM; bar = 100 μm. Comparison analysis of plant height (**e**) and leaf number (**f**) between wild type and *MI*. Error bars indicate SD (** *p* < 0.01; Student’s *t* test).

**Figure 2 ijms-23-07244-f002:**
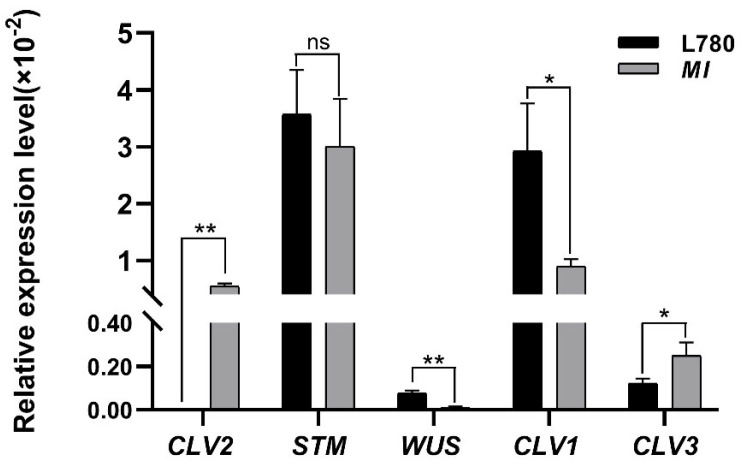
Relative expression of shoot apical meristem (SAM) development-related genes. Error bars indicate SD (* *p* < 0.05, ** *p* < 0.01, ns indicated no significant difference; Student’s *t* test).

**Figure 3 ijms-23-07244-f003:**
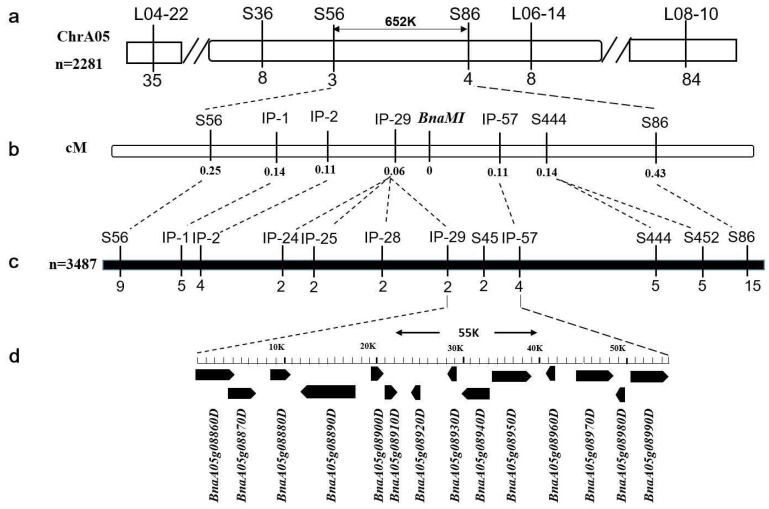
Fine mapping of *BnaMI* gene. (**a**) Initial mapping of *BnaMI* using 2281 BC_3_F_1_ individuals. The applied markers are above the chromosome, and the number of recombinants in this population is marked below. (**b**) The genetic linkage map of *BnaMI*. The values under the markers indicate centimorgan values. (**c**) Fine mapping of *BnaMI* using 3487 BC_4_F_1_ individuals. The markers are arranged according to their physical positions on the chromosome and the numbers below the makers indicate the number of recombinants. (**d**) Annotation in 55 K region according to Darmor (*Bnassica napus* cultivar) reference genome. The black boxes represent the candidate genes predicted in Darmor database.

**Figure 4 ijms-23-07244-f004:**
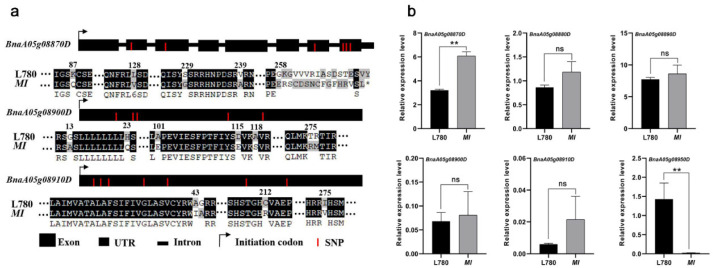
Gene structure, protein alignment, and expression analysis of the candidates. (**a**) Gene structures and variations between alleles from L780 and *MI*. The asterisk symbolizes the terminator, numbers represent the location of amino acids and apostrophes indicate omitted amino acids; (**b**) Relative expression of candidates in shoot apical meristem (SAM). Student’s *t* test was used for statistical analysis; values are the mean ± standard deviation (SD) of three biological replicates (** *p* < 0.01, ns indicated no significant difference).

**Figure 5 ijms-23-07244-f005:**
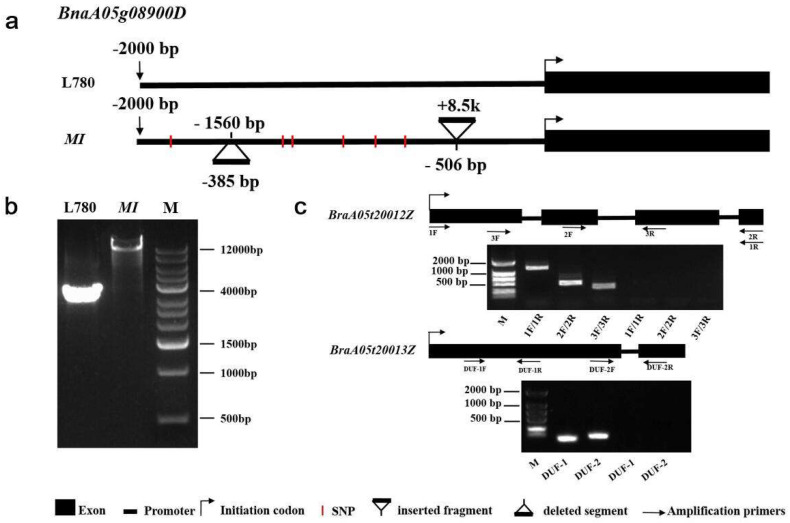
Gene structure, transcript analysis of the large fragment insertion. (**a**) Promoter structures and variations of *BnaA05g08900D* between L780 and *MI*. (**b**) Amplification of large fragment insertion between two mapping parents. (**c**) Transcript analysis of the two genes contained in large insertion. The 1F/1R represents the amplification product of amplification primer 1F, 1R, and DUF−1 represents the amplification product of amplification primers DUF−1F and DUF−1R. F and R represent left and right amplification primers, respectively. M, molecular markers; PCR products were detected by using 1.2% agarose gel.

**Figure 6 ijms-23-07244-f006:**
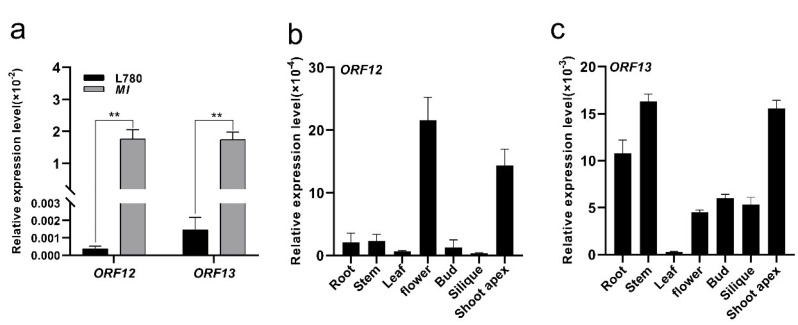
Relative expression and expression pattern of *ORF12* and *ORF13*. (**a**) Relative expression of *ORF12* and *ORF13* in SAM of *MI*. Expression pattern of *ORF12* (**b**) and *ORF13* (**c**) detected by qRT−PCR in roots, stems, leaves, flower, buds, siliques, and shoot apex of *MI*. Values are expressed as average ± SD (n = 3). Student’s *t* test was used for statistical analysis; values are the mean ± standard deviation (SD) of three biological replicates (** *p* < 0.01).

**Table 1 ijms-23-07244-t001:** Annotated information of candidate genes with sequence variation in the 55 kb region.

Gene Number	Arabidopsis Homologous Gene Number	Annotations Information
BnaA05g08870D	AT2G35035.1	urease accessory protein D
BnaA05g08880D	AT2G35030.1	Pentatricopeptide repeat (PPR) superfamily protein
BnaA05g08890D	AT2G35020.1	N-acetylglucosamine-1-phosphate uridylyltransferase 2
BnaA05g08900D	AT2G35000.1	RING/U-box superfamily protein
BnaA05g08910D	AT2G34990.1	RING/U-box superfamily protein
BnaA05g08950D	AT2G34930.1	disease resistance family protein/LRR family protein

## Data Availability

All data generated or analyzed during this study are included in this published article and its Appendix A.

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
