# Peer review of "Identification and Fine Mapping of the Candidate Gene Controlling Multi-Inflorescence in Brassica napus"

_ijms, 2022, doi:10.3390/ijms23137244_

Round 1

Reviewer 1 Report

The composition of the authors' manuscript and the presentation of data are quite lacking in this journal. Of course, discovering new genes through genetic mapping is a difficult study, but it does not seem to fit this journal.

I think the explanation of 1F, 2F, 3F and again 1F, 2F in Fig5C should be explained well in the figure legend or figure.
When we express gene structure in general, shouldn't it be expressed in italics? In almost all figures, BnaA05g08900D, BraA05t20012Z, etc. are not used like that.

Also in Figure 4, while expressing a gene, the gene name or locus number (BnaA05g08870D....) should be in italics, but this was not the case. What's the problem with the authors?

Additionally, if there is too much expression difference, how about expressing the graph separately without cutting the graph?

In Table 1, it is called Arabidopsis homolog of Bna and gene locus number is inserted. Since there are gene names for these, I think it is necessary to use them together. For example At2g35035 is UREASE ACCESSORY PROTEIN D, URED
It would be better to express it as a function that cannot be a name. It seems that the authors misrepresented the names of these genes as functions.

In Figure 2, the full names for CLV, WUS, and STM are already mentioned in the text, but there is no need to use the full names repeatedly. Rather, more information about the experiment should be included in the figure legend.

In Figure 1, the authors made and showed the plant height and the number of leaves as quantified data. Is the number of leaves really around 10 to 15 as shown in the photo? Please show an accurate picture of the plant you counted.

Author Response

Response to Reviewer 1 Comments

Point 1: I think the explanation of 1F, 2F, 3F and again 1F, 2F in Fig5C should be explained well in the figure legend or figure. When we express gene structure in general, shouldn't it be expressed in italics? In almost all figures, BnaA05g08900D, BraA05t20012Z, etc. are not used like that. Also in Figure 4, while expressing a gene, the gene name or locus number (BnaA05g08870D....) should be in italics, but this was not the case.

Response 1: Thank you for your advice. 1F/1R represents the amplification product of amplification primer 1F, 1R, and DUF-1 represents the amplification product of amplification primer DUF-1F, DUF-1R; F and R represents left and right amplification primers, respectively. The clearer descriptions were given for Fig5c. Additionally, the gene names were changed  to italics accordingly. 

Point 2: Additionally, if there is too much expression difference, how about expressing the graph separately without cutting the graph?

Response 2: Thank you for the suggestion. The changes were made in Figure 4b.

Point 3: In Table 1, it is called Arabidopsis homolog of Bna and gene locus number is inserted. Since there are gene names for these, I think it is necessary to use them together. For example At2g35035 is UREASE ACCESSORY PROTEIN D, URED. It would be better to express it as a function that cannot be a name.

Response 3: Thank you for your suggestion. Changes were made in Table 1. For example, ‘’BnaA05g08870D encode an urease accessory protein D, of which the analogs were associated with tillers in rice’’ in manuscript, was change to ‘’BnaA05g08870D was similar to AT2G35035, an urease accessory protein D, of which the analog was  associated with tillers in rice’’.

Point 4: In Figure 2, the full names for CLV, WUS, and STM are already mentioned in the text, but there is no need to use the full names repeatedly. Rather, more information about the experiment should be included in the figure legend.

Response 4: Thank you for your advice. The related issues were revised.

Point 5: In Figure 1, the authors made and showed the plant height and the number of leaves as quantified data. Is the number of leaves really around 10 to 15 as shown in the photo? Please show an accurate picture of the plant you counted.

Response 5: Thank you for your advice. The picture of the seedlings plant of MI mutant and wild-type L780 was provided in supplementary figure S1.

Reviewer 2 Report

Dear editor and colleagues,

I have read the submitted manuscript “Identification and fine mapping of the candidate gene controlling multi‑inflorescence in Brassica napus” with great interest

It is a study that focuses on the fine-mapping of putative genes regulating the multi-inflorescent trait in Brassica napus.

The authors have used appropriate techniques, sampling numbers, and their conclusions are generally supported by data produced.

I have however some comments that the authors could consider in order for their manuscript to be more clear

·         More details regarding the mapping procedure/linkage analysis and data would be beneficial. Report on centimorgan values should be included/indicated [I guess the values under the markers (figure 3) are CM?]. Also include a frequency distribution figure on traits in the BC4F1 population

·         A figure regarding the bp differences across genotypes should be included as a supplementary file.

·         It is not clear why the authors focused on the area between IP-29 and IP-57. Upstream from IP-24 there are also very low recombination events (values approximately ‘2’ are indicated). Perhaps more markers are needed for fine-mapping? Please comment

·         There is a need for English proofreading, since syntax and grammar errors are frequent

Regarding formatting:

·         Abstract and keywords are missing from the manuscript

·         Acronyms should be written in full when first appearing in text. Please check. For instance (L30 SAM). You could include a section delineating abbreviations

·         “Rape” should be changed to “rapeseed “or “oilseed rape” across text

Based on the above I recommend a major revision

Author Response

Response to Reviewer 2 Comments

Point 1: I have read the submitted manuscript “Identification and fine mapping of the candidate gene controlling multi‑inflorescence in Brassica napus” with great interest. It is a study that focuses on the fine-mapping of putative genes regulating the multi-inflorescent trait in Brassica napus. The authors have used appropriate techniques, sampling numbers, and their conclusions are generally supported by data produced.

Response 1: Your constructive suggestions and encouragement for our research were most appreciated. All points raised were addressed as below.

Point 2: More details regarding the mapping procedure/linkage analysis and data would be beneficial. Report on centimorgan values should be included/indicated [I guess the values under the markers (figure 3) are CM?]. Also include a frequency distribution figure on traits in the BC4F1 population

Response 2: Thanks for your suggestions. We have added the centimorgan values in Figure 3b. The values under the markers indicate centimorgan values in Figure 3b. The numbers below the makers in figure 3c indicate the number of recombinants. The frequency distribution figure on traits in BC4F1 population has been added to attachment.

Figure S3. Frequency distribution on multi-inflorescence traits in BC4F1 population. The number above the column represents the actual number of individual plants of multi-inflorescence and single inflorescence respectively.

Point 3: A figure regarding the bp differences across genotypes should be included as a supplementary file.

Response 3: Thanks for your advice. The nucleotide sequence alignment of the candidate genes, including the promoter region and the complete coding region, were provided in supplementary file 2 and supplementary file 3, respectively. To make it clear, each chart contains 4 sequences, the gene sequence of Darmor-bzh (as a reference), the gene sequence of L780 and MI , and the consensus line. In the figure of gene sequence alignment, orange represents SNP, red represents insertion, and blue represents deletion; exons are in black and introns in darkgray. However, the promoter sequence of BnaA05g08890D was not obtained in L780, nor in MI, due to the long N sequence on the reference genome.

Point 4: It is not clear why the authors focused on the area between IP-29 and IP-57. Upstream from IP-24 there are also very low recombination events (values approximately ‘2’ are indicated). Perhaps more markers are needed for fine-mapping? Please comment

Response 4: Thank for the suggestion. We focused on the area between IP-29 and IP-57 according to the number of recombined plants. Although the marker IP-24, IP-25, IP-28 and IP-29 indicated the same number of recombinant plants, the IP-29 marker was located on the far right of the physical location of the reference genome (B. napus Darmor–bzh genome), which implied that candidate gene located on the right side of IP-29 marker. Additionally, the amplification bands of marker S45 were difficult to identify with population screening. In order to ensure the accuracy of mapping, the markers IP-29 and IP-57 were used as flanking markers. Also, we tried to develop SSR markers between IP-24 and IP-57, but no more polymorphic markers were acquired.

Point 5: There is a need for English proofreading, since syntax and grammar errors are frequent.

Response 5: Thank you for your advice. Efforts were made to improve this manuscript, with assistance from a native English speaker.

Point 6: Regarding formatting:

  • Abstract and keywords are missing from the manuscript
  • Acronyms should be written in full when first appearing in text. Please check. For instance (L30 SAM). You could include a section delineating abbreviations
  • “Rape” should be changed to “rapeseed “or “oilseed rape” across text

Response 6: Thank you for your advice. The paper was reviewed and revised accordingly.

Round 2

Reviewer 2 Report

The authors have adequally adressed my coments and also provided new data. 

Acording to my opinion their paper can be accepted as it is